# Long- and Short-Term High-Intensity Interval Training on Lipid Profile and Cardiovascular Disorders in Obese Male Adolescents

**DOI:** 10.3390/children10071180

**Published:** 2023-07-07

**Authors:** Ghazi Racil, Mohamed-Souhaiel Chelly, Jeremy Coquart, Johnny Padulo, Dragos Florin Teodor, Luca Russo

**Affiliations:** 1Research Laboratory (LR23JS01) “Sport Performance, Health & Society”, Higher Institute of Sport and Physical Education of Ksar Said, University of Manouba, Tunis 1000, Tunisia; ghazi_racil@yahoo.fr (G.R.); csouhaiel@yahoo.fr (M.-S.C.); 2Department of Biological Sciences Applied for Physical Activities and Sport, Higher Institute of Sport and Physical Education of Ksar Said, University of Manouba, Manouba 2010, Tunisia; 3Univ. Lille, Univ. Artois, Univ. Littoral Côte d’Opale, ULR 7369-URePSSS-Unité de Recherche Pluridisciplinaire Sport Santé Société, 59000 Lille, France; jeremy.coquart@univ-lille.fr; 4Department of Biomedical Sciences for Health, Università degli Studi di Milano, 20133 Milan, Italy; 5Faculty of Physical Education and Sport, Ovidius University of Constanta, 900029 Constanta, Romania; dragosteodor@yahoo.com; 6Department of Human Sciences, Università Telematica Degli Studi IUL, 50122 Florence, Italy; l.russo@iuline.it

**Keywords:** adolescents, high-intensity interval training, body fat, physical activity, severe obesity

## Abstract

This study investigated the effects of short-term and long-term periods (8 and 16 weeks) of high-intensity interval training (HIIT) on cardiovascular components, blood lipids, and 6-min walking test performance in obese young boys (age = 16.2 ± 0.7) with >34% body fat. The participants were split into two groups: severe obesity (SOG; *n* = 17) and moderate obesity (MOG; *n* = 16). All participants performed on a cycle ergometer for 16 weeks (3 times per week) of HIIT at 100% peak power output at the ventilatory threshold and recovered at 50% of peak power. Except for BMI, both groups improved all body composition measures after 16 weeks, with a higher percentage of change (Δ) in SOG. The 6-min walking test increased in both groups (*p* < 0.001). Furthermore, cardiovascular variables, blood lactate concentration at rest and after 5-min post-exercise, blood lipids, and insulin concentrations improved significantly in both groups. After 16 weeks, MOG significantly improved in HR_peak_, blood glucose concentration, and rating of perceived exertion (RPE), but the percentage of change (Δ) was higher in SOG for all the other variables. SOG showed a higher (Δ) waist-to-hip ratio, maximum heart rate, resting heart rate, systolic blood pressure, blood lactate at 5-min post-exercise, and triglyceride concentrations after 8 and 16 weeks of training. In conclusion, a long-term HIIT program appears to be an appropriate training approach for obese boys with extra body fat. However, considering the RPE values, short-duration training sessions should be planned.

## 1. Introduction

Obesity rates continue to rise worldwide, reflecting a widespread condition characterized by excessive accumulation of body fat in all age groups and genders. As such, this increasing prevalence of obesity leads to a range of health complications [1,2], including the potential development of insulin resistance in children [3,4], leading to a disturbance in glucose metabolism [5], and contributing to the development of hypertension [6,7,8]. Research indicates that approximately half of adults with hypertension experienced elevated blood pressure levels during their childhood, largely attributed to insufficient physical activity [9]. Although the optimal exercise program specifically adapted to this population sometimes remains undefined in certain cases, physical exercise remains an effective intervention to counteract pediatric obesity [10,11,12].

However, when individuals engage in long-term training at moderate intensity (at 40–50% of maximum power), this leads to a reduction in body fat, improves its oxidation, and therefore acts on insulin concentration levels [13], making it well tolerated by people suffering from obesity [14]. However, when individuals engage in long-duration training at moderate intensity (at 40–50% of maximum power), this leads to the oxidation of body fat and thus decreases insulin concentration [13], which makes it tolerable for people suffering from obesity [14]. On the other hand, high-intensity interval training (HIIT), which creates a negative energy balance, also reduces adiposity in overweight and obese people [15] and improves health status [16]. Several research studies have highlighted the remarkable physiological benefits and positive impact of HIIT on cardio-metabolic risk factors [17,18] and insulin resistance [19,20] in obese young people. This training method also helped increase high-density lipoprotein cholesterol (HDL-C) levels [16,21,22], while reducing overall blood cholesterol concentration [16,23] and improving glucose tolerance [24,25].

In general, the presence of body fat is an indication of health risks in humans [26,27], while its significant reduction positively impacts health. A moderately obese person has a body mass index (BMI) between 30.0 and 34.9 kg·m^−2^, while a severely obese person has a BMI between 35.0 and 39.9 kg·m^−2^ [28]. However, it remains essential for obese people to reduce their weight, waist circumference, and body fat. All these factors can be improved by anaerobic exercise, including HIIT, but they result in high blood lactate production [29].

Although studies indicate that short-term high-intensity interval training (ST-HIIT) interventions can reduce body mass in obese boys, this reduction has not been maintained over an extended period [30]. In their study, Milanović et al. [31] demonstrated that a 12-week HIIT program significantly improved cardiorespiratory fitness, body composition, and insulin sensitivity in obese individuals. Similar results, however, have been reported in another research [25,32]. Even though low-volume HIIT has been shown to be effective in reducing body fat during both short- and long-term training periods [33,34], to our knowledge, no specific studies have been dedicated to discovering the effects of such a type of training in subjects with severe obesity.

Under such circumstances, it becomes interesting to explore the effect of body fat levels on obese youth in different obesity classes after performing the HIIT mode for short- or long-term training periods.

This study therefore aims to investigate the effects of HIIT (at 100% maximum power at ventilatory threshold) after short and long training periods on boys with moderate or severe obesity on cardiovascular variables, lipid profiles, and blood lactate concentrations and to evaluate their perceived exertion at the completion of the effort.

We hypothesize that boys with moderate obesity will exhibit more significant enhancements in cardiovascular and biological parameters and demonstrate better adaptation to HIIT in both the short and long term. Conversely, boys with severe obesity may require longer training periods exceeding 8 weeks to adapt effectively to HIIT, but they may struggle to adhere to prolonged and intense training programs.

## 2. Materials and Methods

### 2.1. Participants

All the participants (Table 1) underwent only two hours per week of physical activity as part of the school’s physical education program. None of the boys participated in any systematic exercise training in the last three months before the study’s commencement and were asked to abstain from any forms of aerobic or anaerobic physical activity other than their school’s physical education program. None of the participants used drugs or therapies for obesity or presented with chronic diseases. All boys were asked not to consume any medications during the week prior to blood sampling, which may have impacted the study’s testing and progress. Prior to participation in the study, parents received a complete verbal description of the study, outlining the protocol and potential risks and benefits. Thereafter, written informed consent was obtained from all parents. Furthermore, this study was conducted in accordance with the Declaration of Helsinki. This study was approved by the local Research Ethics Committee under the number (No. 241/2022).

Participants’ sampling was based on three local school-based recruitments through consultations with specialized pediatricians. Participants with a BMI ≥ 97th percentile according to French standards and a percentage of body fat (%BF) ≥ 34% [35] were accepted for this study and later classified according to their body mass index (BMI) class. In total, thirty-five participants were recruited for the study (age = 16.2 ± 0.7 years) and were assigned to moderately (MOG, *n* = 18) or severely (SOG, *n* = 17) obese groups (see Table 1 for detailed data on the sample).

### 2.2. Anthropometrical Measurements

On the first day, body mass (BM) to the nearest 0.1 kg and lean body mass (LBM) were assessed with a digital scale (Tanita, Tokyo, Japan) [16]. The body height was measured in cm with the participant scantily dressed and without shoes as standard procedure (Model 214 height rod; Seca, Hamburg, Germany). Body mass index (BMI) was calculated using the algorithm provided by the Centers for Disease Control and Prevention (CDC) (body mass [kg])/(body height [m])^−2^. Percent body fat (%BF) was assessed for each participant by bioelectrical impedance analysis (BIA) (TBF-300, Tanita, Tokyo, Japan). As recommended by international guidelines and at the end of gentle-expiration, waist circumference (WC) and Hip circumference in cm were measured with a non-deformable tape ruler with participants standing and breathing normally. Thereafter, the Waist-hip ratio (WHR) was calculated. Abdominal obesity is defined as a WHR above 0.90 for males [36]. All participants were instructed to maintain their usual diet and lifestyle.

During the same trial, a graded exercise test on an electromagnetically braked cycle ergometer was performed (Ergometrics 800, Ergoline^®^, Blitz, Germany). Initial power output was set at 10 W, increasing by 10 W·min^−1^ until volitional exhaustion (pedaling rate fixed to 60 revolutions per minute). Heart rate (HR) was continuously recorded using a 12-lead electrocardiogram (Medcard, Medisoft^®^, Sorinnes, Belgium), while alveolar gas exchanges were recorded using an ergo-spirometry device (Ergocard, Medisoft^®^, Sorinnes, Belgium). The v-slope method of Beaver et al. [37] was used to identify the ventilatory threshold (VT) and the power output at VT.

On the second day, the maximal distance performed during a 6-min walking test (6-min WT) was measured on an outside 200 m running track. Before the test’s commencement, the test procedure was explained to participants. They were asked to walk as far as possible within 6-min, without running. Participants could stop at any moment and as much as they needed to.

### 2.3. Blood Measurements

On the third day and after twelve hours of fasting, cholesterol (TC), triglycerides (TG), and HDL-C levels were measured for all participants using enzymatic methods. The same tests were repeated after 8 and 16 weeks under the same conditions. The inter-assay coefficients of variation (CV) were 1.7, 2.2, and 2.0%, respectively. LDL-C was calculated using the formula by Friedewald et al. [38]. Plasma glucose concentration was measured by the hexokinase method using an automated device (Architect c8000, Abbott^®^, Quebec, QC, Canada), and 1.7% was the inter-assay CV. Insulin concentration was assayed by an IRMA insulin kit (Immunotech^®^, Marseille, France). The intra- and inter-assay coefficients of variation were 3.3–4 and 3.7–4.8%, respectively.

An estimate of insulin resistance was calculated by the homeostasis model assessment of insulin resistance index (HOMA-IR). The blood lactate (BL) samples were collected at rest and immediately 5-min (BL_5 min_) after the completion of the last training session in each testing period. The lactate levels were measured from the capillary blood samples using the portable device “Lactate Scout4” (SensLabGmbH, Leipzig, Germany) [39].

The resting systolic and diastolic blood pressures (SBP and DBP in mmHg, respectively) and the resting heart rate (HR_rest_ in b·m^−1^) were monitored (after sitting for 10-min) using an arm tensiometer (Spengler^®^, Issoudun, France) and a heart rate monitor (S-610, Polar^®^, Kempele, Finland), respectively. The peak heart rate (HR_peak_) was also defined.

### 2.4. Training Programs

The two training groups followed a 16-week training program, which consisted of three sessions of intermittent exercise per week (i.e., Tuesday, Thursday, and Saturday) on a cycle ergometer (Ergometrics 800, Ergoline^®^, Blitz, Germany). Each session began with a 5-min standardized warm-up and ended with a 5-min cool-down performing static stretching. Participants performed 20-min of HIIT consisting of two blocks of 10 bouts of 30-s cycling at 100% Peak power output (PPO) at the ventilatory threshold interspaced by 30-s recovery cycling at 50% PPO (i.e., close to the intensity eliciting maximal fat oxidation rate) [40]. The two blocks were separated by a 5-min passive recovery period. Therefore, a progressive increase was devoted to the total training session duration, which increased from 35 to 45 and to 55 min during the first four weeks, the second four weeks, and the remaining eight weeks, respectively. Each continuous four weeks, the exercise intensity could be updated (i.e., 5% power output increased when the HR at the ventilatory threshold decreased by 5 b·m^−1^).

At the first training session, the last training session of the eighth week, and the last training session of the sixteenth week, all participants were asked to rate RPE from the French translation [41] of the Foster et al. scale [42], thirty minutes after the training session had ended.

The tests took place in the morning after a standardized breakfast and under similar environmental conditions, as usual for field and laboratory tests [43,44,45,46,47], to reduce any circadian effect [48]. The breakfast was composed of (10 kcal·kg^−1^, ~55% by carbohydrates, ~33% by lipids, and ~12% by proteins, as defined by an experienced nutritionist) and under similar environmental factors. All participants were instructed to maintain their usual physical activity level and their usual diet during the intervention period and not to consume any medications during the week prior to blood sampling. In total, five adolescents dropped out during the training period for personal reasons (2 from MOG and 3 from SOG), and thus they were not accounted for in the groups and their results were excluded.

### 2.5. Statistical Analysis

Descriptive data are presented as the mean and standard deviation (SD). Shapiro-Wilk and Levene’s tests were used to verify the normal distribution of the data and the homogeneity of variances, respectively. An analysis of variance (ANOVA) was used to compare baseline characteristics among the groups.

A two-way analysis of variance (ANOVA) with repeated measures was used to compare all variables within each group at the three data collection points to determine the main effect and interaction effect on the different variables. Inter-class correlation coefficients (ICC) were calculated to evaluate the reliability of body composition. Upon detection of a significant interaction effect, Tukey’s honest significant difference test was applied to compare all possible pairs of groups. The effect size (ESs) for the time × group interaction effects was calculated for all ANOVAs using partial eta-squared ($\eta_{p}^{2}$). Effect sizes (ES) were determined by converting the partial eta-squared from the ANOVA output to Cohen’s *d*. The ES enabled estimating the magnitude of the difference (i.e., trivial: ES < 0.2, small: 0.2 ≤ ES < 0.5, moderate: 0.5 ≤ ES < 0.8, and large: ES ≥ 0.8). The level of significance was set at *p* < 0.05. Between-group changes were performed by Friedman to compare all times. In cases of significance, the Wilcoxon test was applied to locate the difference between groups (T0–T1, T0–T2, and T1–T2). Statistical analysis was performed using SPSS (IBM SPSS Statistics for Windows, Version 24.0. Armonk, NY, USA: IBM Corp.).

## 3. Results

Anthropometric and cardiopulmonary data before and after the intervention programs are shown in Table 1. For all body composition measurements, the reliability was excellent (ICC > 95%). Comparisons of the two groups before the intervention showed that they were matched for age and anthropometric parameters. At post-intervention, significant improvements were shown (Table 1) in severe and moderate obesity groups for BM and BMI (*p* < 0.01), %BF, LBM, RPE, and the 6-min walking test (*p* ˂ 0.001). Furthermore, at post-intervention, WC and HipC decreased significantly (*p* < 0.01, ES = 0.75 and ES = 0.57 against ES = 0.69 and ES = 0.52, in SOG and MOG, respectively). Although the WHR decreased significantly (*p* < 0.05) in both groups, the SOG presented the higher percentage of change after 16 weeks (−1.93% against −1.34% in MOG).

In Table 2, significant improvements are shown in both groups at the intervention end (*p* < 0.01) in all: HR_peak_, DBP, and HDL-C (ES = 0.49, ES = 0.54, and ES = 0.68 against ES = 0.54, ES = 0.42, and ES = 0.66; in SOG and MOG, respectively), whereas for the blood glucose, MOG exhibited a higher decrease, ES = 0.69 compared to the SOG, ES = 0.59, and was significantly lower in the between groups’ comparison in the post-intervention. The insulin and HOMA-IR also decreased (*p* < 0.001). Indeed, significant exercise-related decreases were noted in BL5 min (*p* < 0.01). From another side, SOG experienced a higher significant improvement compared to MOG in all: RHR (*p* < 0.01 against *p* < 0.05, respectively), SBP, resting blood lactate, and TC (*p* < 0.001 against *p* < 0.05), whilst for TG and LDL-C, the decreases were significant at (*p* < 0.001 against *p* < 0.01, respectively).

In the between-groups’ comparison, MOG showed significant improvements in post-intervention in BL_rest_ and BL_5 min_, glucose, TC, TG, LDL-C, and HOMA-IR values (*p* = 0.032, ES = 0.59; *p* = 0.044, *p* = 0.012, *p* = 0.017, *p* = 0.038, *p* = 0.014, and *p* = 0.027, respectively). After 30 min of the training session ending, both groups showed a reduction in RPE that corresponded with the various periods (pre- vs. post-8 weeks; pre- vs. post-16 weeks; and post-8 vs. post-16 weeks: −18.18%, −17.04%, and 1.38%, against −20.0%, −21.17%, and −1.47%, respectively). The results of the between-group comparison are shown in Table 3 and Table 4.

## 4. Discussion

The main goal of this study was to compare the effects of ST-HIIT and LT-HIIT on body composition, cardiovascular factors, and blood lipids in adolescent boys with severe and moderate obesity. The findings show that in boys with a severe level of obesity, most of the blood lipids, cardio-respiratory fitness, body composition, and HOMA-IR improved more after LT-HIIT. However, after ST-HIIT (8 weeks), the MOG improved more in the body composition variables and was able to perceive the effort as less intensive during all the testing periods compared to the SOG.

After a vigorous workout, the muscle fibers become activated, resulting in increased stimulation and strength in the muscles, enabling practitioners to tap into their fat reserves and use them as an energy source [49]. Despite what has been reported, no effect was detected on lean mass or fat mass in the trunk or abdomen of overweight or obese young women after following a five-week HIIT program [50]. Similarly, Mendelson et al. [51] failed to show a decrease in fat mass in sixty overweight or obese subjects after a short period of HIIT. The authors suggested that this could be a consequence of the short intervention period compared with the studies, which used at least 12 weeks of training [52]. In the current study, whether it was after 8 weeks or 16 weeks, the SOG experienced the biggest decrease in WC and WHR. This has a substantial impact since WHR provides us with information on the distribution of fat tissue inside the body, which we suppose accurately measures the cardiometabolic risk connected with obesity in young obese individuals [53].

Moreover, our collected results were able to demonstrate that both ST-HIIT and LT-HIIT considerably increased aerobic capacity through significant improvements in the 6-min walking test. This was supposed to be related to the intense exercises having lowered body fat, improving the muscles’ ability to tolerate stress better [54], which is supposed to lower the risk of cardiovascular disease [55,56].

Both groups have further experienced large decreases in HR_peak_ and RHR, which may have helped them feel less worn out and exhausted while cycling. According to the literature, a high intensity of 90–95% of an individual’s maximum heart rate is regarded as a prerequisite for successful adaptations in endurance performance [57]. According to Ratel et al. [58], young people prefer to take part in short bursts of intense physical activity followed by periods of lower intensity [59].

Post-intervention, both groups significantly reduced blood glucose levels (see Table 4). These enhancements are assumed to be related to the training intensity, thereby helping in the development of GLUT4 protein content and leading to an effective glycogen supercompensation [60]. In fact, high-intensity exercise necessitates glucose catabolism, and as a result, glucose metabolism is turned away to replenish muscle glycogen deposits [61]. According to Host et al. [62], this reduction in glucose may also be attributed to improvements in insulin sensitivity. It should be noted that obese people who have a low uptake of oxygen and glucose in the muscle [63,64], when following high-intensity exercise, can turn away glucose metabolism to restore muscle glycogen stores, which are primarily resynthesized from muscle triacylglycerols during recovery [65]. Whether it was after 8 or 16 weeks, both groups improved TC, TG, LDL-C, and HDL-C, but the larger percentage of change was noted in SOG after 16 weeks, confirming the positive contribution of such training in obese people, whatever their level of body fat concentration. On the other hand, it is supposed that the drop in LDL-C in both training groups may be related to the improvement in total fat metabolism. This is congruent with the findings of Racil et al. [25]. From their side, Kong et al. [60] noted that a five-week HIIT program has improved glucose metabolism in obese young women, similar to what was reported in both groups, but contradicts another study, which examined the impact of 8 weeks of HIIT on body composition and lipid profile in overweight teenagers and found significant variations for HDL-C and TC values but not for TG and LDL-C despite significant differences in weight, BMI, and body fat percentage [66].

It is worth mentioning that both groups significantly decreased their SBP and DBP after 8 and 16 weeks of training, which is supposed to be related to the training mode and the exercise intensity. Generally, obese subjects present hypertension in childhood [67], and when we intervene at the right time, we can improve the quality of health and well-being of the person. In contrast, Popowczak et al. [68] showed that 10-weeks of HIIT could significantly reduce SBP among high blood pressure adolescents but had no effect on DBP. We believe that changes in blood vessel shape and function to increase blood flow [69] may explain the larger decrease in blood pressure observed after 16 weeks of training in favor of the SOG. This may help improve the lessened compliance of the essential elastic arteries, lowering long-term arteriole vasoconstriction [70]. On the other hand, the current study’s participants decreased WC by more than 3 cm, which we suggest may improve metabolic indicators [71] and may improve the components of the metabolic syndrome [72].

Following the biological analysis, the SOG showed a higher percentage of change (−34.10 ± 6.3%) compared to the MOG (−31.86 ± 7.2%). This may result from decreasing the WHR, which is considered an important element for detecting a health-related complication [53]. This finding is consistent with the findings of 12- or 16-week HIIT programs in obese adolescents [25,32], where insulin was improved favorably.

In most cases, obese people present high values of insulin concentration at rest [73], and when a subject presents a value of 15 μU·mL^−1^ blood insulin levels, they are regarded as hyper-insulinemic [74]. Before the intervention, both groups noted a level of HOMA-IR more than 3.16, which is considered the high limit index [75]. After 8 weeks, only MOG decreased the HOMA-IR below that value. Even though the SOG decreased its value to 2.73 ± 3.52 post-intervention, the MOG value remained the lowest (2.44 ± 4.6), validating the suitability of training duration and exercise intensity.

Regarding the BL_5 min_ levels, there was a considerable reduction, whether it was at ST- or LT-HIIT periods. Concerning the resting blood lactate level, SOG showed the biggest decrease during the second period of the intervention. It is worth noting that both groups had an excess of %BF at the start, which usually prevents obese people from recovering quickly after exercise and lowers their lactate levels [76]. These reported decreases could be related to changes in muscle fiber type or to glycogen storage capacity in the muscles after training. We believe, therefore, that it would be better to repeat training sessions at high intensity levels, which may condition adolescents to strain their glycolytic metabolism, allowing for progressively stronger lactic acid buffering.

As previously demonstrated [77], intra-individual comparison of perceptual values (here measured by RPE) can be utilized to assess the influence of exercise training programs on physical fitness in obese participants. In the current study, an 8-week training program was adequate to reduce RPE by 1.6 units in the SOG group and 1.7 units in the MOG. As a result, the percentage change in the MOG was substantially higher than in the SOG. In fact, the MOG lowered its level with no significant change between the eighth and sixteenth weeks, with a percentage change of (−1.47%), whereas the SOG’s RPE value climbed with (1.38%). This result appears to be mostly due to an increase in the session’s training length, which became 55-min compared to 45-min until the eighth week. We were not expecting that the length of the training session would influence the subject’s perception of difficulty even after the completion of the training period. Therefore, we believe that the feelings that emerged during LT-HIIT do not encourage severely obese people to adhere to long and intense training programs. To counteract this situation, it is possible that HIIT associated with moderate-intensity exercises in severely obese individuals would provide higher benefits than only HIIT.

To be more accurate, the results collected from this study can provide valuable practical insights for teachers working with obese adolescents in designing effective and safe exercise regimens to educate their students about potential risks, such as overexertion or joint stress, and teach them proper form, warm-up routines, and cool-down exercises to reduce the likelihood of injuries. On the other hand, teachers can use this type of training and respect the exercise and program duration knowledge to develop tailored programs that suit the specific needs of obese adolescents, considering their current fitness level. Furthermore, teachers should consider starting with low- to moderate-intensity exercises and gradually increasing intensity levels as the obese adolescents’ fitness improves over time. Consequently, by monitoring heart rate and perceived exertion during training sessions, teachers can educate and train obese adolescents to gauge their intensity levels based on these measures, ensuring that they exercise within a safe and appropriate range. Therefore, to broaden the generalizability of findings, future research could include a more diverse participant sample, such as both boys and girls or participants from diverse cultural backgrounds. Indeed, studies that involve both obese and non-obese participants would also provide valuable insights. Since this investigation lacked an adaptation period, future research could incorporate a pre-intervention phase where participants gradually adapt to the exercise program.

### Limitations

Some limitations of this study are acknowledged. Firstly, the sample included only boys, so further experiments must be conducted to confirm the collected results in a larger and more varied population. On the other hand, one must keep in mind that the changes observed in the current study could be due in part to the participants’ lower baseline fitness levels, as rapid improvements are more likely to occur in sedentary individuals [78].

## 5. Conclusions

The current study’s results let us conclude that ST and LT-HIIT have been shown to be effective and sufficient modes of exercise to improve body composition and cardiovascular and metabolic abnormalities in moderately and severely obese young males. Given that severely obese boys rated workout intensity highly in long-term sessions, it seems better to start by gradually increasing exercise duration and intensity over time, as this may be more appropriate and thus avoid limited exercise tolerance and a high risk of injury. Consequently, further research in this context is needed to confirm these findings and determine the generalizability of the results to other populations.

## Figures and Tables

**Table 1 children-10-01180-t001:** Body composition and cardiopulmonary data (mean ± SD), before and after 8 and 16 weeks of the intervention periods.

	SOG (*n* = 17)				MOG (*n* = 18)			
Variables	Pre	Post 8w	Post 16w	ES	Pre	Post 8w	Post 16w	ES
Age (years)	16.0 ± 0.5		16.3 ± 0.4		16.4 ± 0.6		16.7 ± 0.6	
Height (cm)	168.0 ± 4.2	168.0 ± 4.4	16.3 ± 0.4	NS	165.6 ± 3.0	165.6 ± 3.4	165.7 ± 4.2	NS
BM (kg)	103.1 ± 5.2	101.1 ± 5.0 ^a^	97.4 ± 3.4 ^b1^	0.57	86.7 ± 4.2	84.1 ± 3.7 ^a^	82.3 ± 3.3 ^b1, £^	0.54
BMI (kg·m^−2^)	36.56 ± 1.6	35.85 ± 1.5 ^a^	34.52± 4.5 ^b1^	0.68	31.8 ± 1.6	30.1 ± 1.3 ^a^	29.2 ± 3.5 ^b1, £^	0.72
BF (%)	45.6 ± 4.1	42.4 ± 4.3 ^a^	40.6 ± 4.7 ^c1^	0.67	41.9 ± 4.3	39.2 ± 3.2 ^b^	37.6 ± 4.2 ^c1, £^	0.63
LBM (kg)	51.2 ± 2.1	54.4 ± 3.3 ^b^	57.7 ± 3.2 ^c1^	0.72	50.7 ± 3.1	53.9 ± 2.7 ^b^	56.1 ± 2.1 ^c1^	0.54
WC (cm)	98.8 ± 8.2	96.5 ± 6.6 ^a^	93.3 ± 6.1 ^b1^	0.75	94.4 ± 7.1	92.7 ± 6.9 ^a^	91.1 ± 4.9 ^a1, £^	0.69
HipC (cm)	99.9 ± 9.3	99.2 ± 7.2	96.2 ± 8.3 ^b1^	0.57	96.3 ± 7.5	95.5 ± 8.6	94.2 ± 6.7 ^a1^	0.52
WHR (%)	0.98 ± 0.03	0.97 ± 0.03 ^a^	0.96 ± 0.04 ^a1^	0.54	0.98 ± 0.04	0.97 ± 0.03	0.96 ± 0.04 ^a1^	0.51
6 min WT (m)	582 ± 61.2	612 ± 71.6 ^b^	636 ± 72.5 ^c1^	0.65	594 ± 72.4	625 ± 67.4 ^b^	646 ± 69.4 ^c1^	0.59
RPE (a.u.)	8.8 ± 0.7	7.2 ± 0.9 ^c^	7.3 ± 0.8 ^c1^	0.59	8.5 ± 0.5	6.8 ± 0.5 ^c^	6.7 ± 0.6 ^c1^	0.67

Note: Significantly different with pre-test: “^a^”, “^a1^” at *p* < 0.05; “^b^”, “^b1^” at *p* < 0.01; “^c^”, “^c1^” at *p* < 0.001; “^£^” Significantly different from the other group in post-intervention at *p* ˂ 0.05. SOG: severe obesity group; MOG: moderate obesity group; BM: body mass; BMI: body mass index; BF (%): percentage of body fat; LBM: lean body mass; WC: waist circumference; HipC: hip circumference; WHR: waist to hip ratio; RPE: rating of perceived exertion; ES: effect size.

**Table 2 children-10-01180-t002:** Physiological and biological data before and after 8 and 16 weeks of the intervention periods at HIIT.

	SOG (*n* = 17)				MOG (*n* = 18)			
Variables	Pre	Post 8w	Post 16w	ES	Pre	Post 8w	Post 16w	ES
HR_peak_ (b·m^−1^)	197.6 ± 13.8	188.9 ± 12.6 ^b^	188.1 ± 11.7 ^b1^	0.49	195.2 ± 14.4	186.9 ± 12.6 ^b^	185.4 ± 12.6 ^b1^	0.54
RHR (b·m^−1^)	73.2 ± 3.0	70.4 ± 2.0 ^a^	69.5 ± 21.0 ^b1^	0.51	70.4 ± 2.0	68.6 ± 4.0 ^a^	68.5 ± 2.2 ^a1^	0.47
SBP (mm·Hg^−1^)	129 ± 8.0	121.2 ± 5.0 ^b^	118.2 ± 4.1 ^c1^	0.72	119 ± 6.0	117 ± 5.0 ^a^	116.6 ± 4.4 ^a1^	0.53
DBP (mm·Hg^−1^)	84 ± 4.0	82.2 ± 3.0 ^a^	80.7 ± 2.2 ^b1^	0.54	83 ± 3.0	81.2 ± 2.0 ^a^	80.2 ± 2.3 ^b1^	0.42
BL_rest_ (mmol·L^−1^)	1.4 ± 0.4	1.38 ± 0.3	1.31 ± 0.5 ^c1^	0.72	1.3 ± 0.3	1.27 ± 0.2 ^a^	1.26 ± 0.7 ^a1, £^	0.67
BL_5 min_ (mmol·L^−1^)	15.2 ± 1.5	12.4 ± 1.4 ^c^	11.4 ± 2.6 ^c1^	0.75	14.6 ± 1.3	12.4 ± 1.3 ^a^	11.07 ± 1.3 ^c1, £^	0.72
Glucose_rest_ (mmol·L^−1^)	4.63 ± 0.11	4.44 ± 0.13 ^a^	4.3 ± 0.14 ^b1^	0.59	4.40 ± 0.17	4.05 ± 0.16 ^b^	3.95 ± 0.16 ^b1, £^	0.69
TChol (mmol·L^−1^)	4.4 ± 0.25	4.2 ± 0.31 ^a^	3.9 ± 0.27 ^c1^	0.76	3.8 ± 0.21	3.6 ± 0.32 ^a^	3.64 ± 0.29 ^a1, £^	0.62
TG (mmol·L^−1^)	1.6 ± 0.05	1.5 ± 0.06 ^b^	1.4 ± 0.07 ^c1^	0.64	1.48 ± 0.05	1.4 ± 0.06 ^b^	1.35 ± 0.09 ^b1, £^	0.49
HDL-C (mmol·L^−1^)	0.94 ± 0.05	0.96 ± 0.05 ^b^	1.0± 0.07 ^b1^	0.68	0.97 ± 0.08	0.99 ± 0.05 ^a^	1.03 ± 0.06 ^b1^	0.66
LDL-C (mmol·L^−1^)	3.14 ± 0.27	2.94 ± 0.3 ^b^	2.62 ± 0.32 ^c1^	0.62	2.53 ± 0.4	2.33 ± 0.3 ^b^	2.34 ± 0.25 ^b1, £^	0.46
Insulin (μU·mL^−1^)	21.7 ± 1.8	16.2 ± 1.3 ^c^	14.3 ± 1.6 ^c1^	0.72	20.4 ± 1.5	15.2 ± 0.2 ^c^	13.9 ± 1.9 ^c1^	0.68
HOMA-IR	4.46 ± 0.52	3.19 ± 0.63 ^c^	2.73 ± 3.52 ^c1^	0.69	3.98 ± 0.46	2.74 ± 0.42 ^c^	2.44 ± 4.6 ^c1, £^	0.71

Note: Significantly different with pre-test: “^a^”, “^a1^” at *p* < 0.05; “^b^”, “^b1^” at *p* < 0.01; “^c^”, “^c1^” at *p* < 0.001; “^£^” Significantly different from the other group in post-intervention at *p* ˂ 0.05. SOG: severe obesity group; MOG: moderate obesity group; HR_peak_: heart rate peak; RHR: resting heart rate; SBP: systolic blood pressure; DBP: diastolic blood pressure; BL_rest_: blood lactate concentration at rest; BL_5 min_: blood lactate concentration 5 min after the exercise ends; TChol: total cholesterol; TG: triglyceride; HDL-C: high-density lipoprotein cholesterol; LDL-C: low-density lipoprotein cholesterol; HOMA-IR: homoeostasis model assessment index for insulin resistance; ES: effect size.

**Table 3 children-10-01180-t003:** Comparison of the correspondent percentage changes at three times between SOG and MOG in Body composition and cardiopulmonary data.

	SOG (*n* = 17) % Change				MOG (*n* = 18) % Change			
Variables	Pre vs. Post 8w	Pre vs. Post 16w	Post 8w vs. Post 16w	ES	Pre vs. Post 8w	Pre vs. Post 16w	Post 8w vs. Post 16w	ES
BM (kg)	−1.93	−5.52 ^a^	−3.65 ^a^	0.48	−2.99 ^aa^	−5.07	−2.14	0.42
BMI (kg·m^−2^)	−1.94	−5.57	−3.70	0.46	−3.00 ^aa^	−7.71 ^aa^	−4.85 ^aa^	0.61
BF (%)	−2.95	−9.85 ^a^	−7.10 ^a^	0.46	−7.52 ^bb^	−9.19	−1.80	0.42
LBM (kg)	6.25	12.69 ^a^	6.06 ^a^	0.58	6.31	10.65	4.08	0.48
WC (cm)	−2.32 ^a^	−5.56 ^b^	−3.31 ^a^	0.61	−1.80	−3.49	−1.72	0.56
HipC (cm)	−0.70	−3.70 ^b^	−3.02 ^a^	0.47	−0.83	−2.18	−1.36	0.44
WHR (%)	−1.63 ^a^	−1.93 ^b^	−0.30	0.52	−0.97	−1.34	−0.36	0.47
6 min WT (m)	5.15	9.27 ^a^	3.92 ^a^	0.69	5.21	8.75	3.36	0.57

Note: Significantly different from MOG in the same corresponding training period: “^a^” at *p* < 0.05; “^b^” at *p* < 0.01; Significantly different from SOG in the same corresponding training period: “^aa^” at *p* < 0.05; “^bb^” at *p* < 0.01; ES: effect size.

**Table 4 children-10-01180-t004:** Comparison of the correspondent percentage changes at three times between SOG and MOG on Physiological and biological data.

	SOG (*n* = 17) % Change				MOG (*n* = 18) % Change			
Variables	Pre vs. Post 8w	Pre vs. Post 16w	Post 8w vs. Post 16w	ES	Pre vs. Post 8w	Pre vs. Post 16w	Post 8w vs. Post 16w	ES
HR_peak_ (b·m^−1^)	−4.40 ^a^	−4.80	−0.42	0.47	−4.25	−5.02 ^aa^	−0.80 ^aa^	0.53
RHR (b·m^−1^)	−3.82 ^a^	−5.05 ^b^	−1.27 ^a^	0.52	−2.55	−2.69	−0.14	0.47
SBP (mm·Hg^−1^)	−6.04 ^c^	−8.37 ^c^	−2.47 ^a^	0.57	−1.68	−2.01	−0.34	0.37
DBP (mm·Hg^−1^)	−2.14	−3.92 ^a^	−1.82 ^a^	0.48	−2.16	−3.37	−1.23	0.42
BL_rest_ (mmol·L^−1^)	−1.42 ^a^	−6.42 ^b^	−5.07 ^a^	0.53	−2.30	−3.07	−0.78	0.44
BL_5 min_ (mmol·L^−1^)	−18.42 ^a^	−25.0 ^a^	−8.06	0.76	−15.06	−24.17	−10.72 ^aa^	0.61
Glucose_rest_ (mmol·L^−1^)	−4.10	−7.12	−3.15 ^a^	0.53	−7.95 ^aa^	−10.22 ^cc^	−2.46	0.64
TChol (mmol·L^−1^)	−4.54	−11.36 ^c^	−7.14 ^c^	0.64	−5.26 ^a^	−4.21	1.11	0.51
TG (mmol·L^−1^)	−6.25 ^a^	−12.5 ^b^	−6.66 ^b^	0.51	−5.40	−8.78	−3.57	0.42
HDL-C (mmol·L^−1^)	2.12	6.38 ^a^	4.16	0.58	2.06	6.18	4.04	0.52
LDL-C (mmol·L^−1^)	−6.36	−16.56 ^c^	−10.88 ^c^	0.52	−8.05 ^aa^	−7.65	0.42	0.41
Insulin (μUml^−1^)	−25.34	−34.10 ^b^	−11.72 ^b^	0.72	−25.49	−31.86	−8.55	0.67
HOMA-IR	−28.40	−38.79	−14.51 ^b^	0.66	−31.41 ^aa^	−38.83	−10.81	0.68
RPE (a.u.)	−18.18	−17.04	1.38	0.64	−20.0 ^aa^	−21.17 ^bb^	−1.47 ^aa^	0.71

Note: Significantly different from MOG in the same corresponding training period: “^a^” at *p* < 0.05; “^b^” at *p* < 0.01; “^c^” at *p* < 0.001; Significantly different from SOG in the same corresponding training period: “^aa^” at *p* < 0.05; “^bb^” at *p* < 0.01; “^cc^” at *p* < 0.001; ES: effect size.

## Data Availability

The data that support the findings of this study are available from the corresponding author upon reasonable request.

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
