# Peer review of "Long- and Short-Term High-Intensity Interval Training on Lipid Profile and Cardiovascular Disorders in Obese Male Adolescents"

_children, 2023, doi:10.3390/children10071180_

Round 1

Reviewer 1 Report

Given that the global prevalence of obesity in children and adolescents has now reached alarming levels, defined by the World Health Organization (WHO) as a "global epidemic escalation", it is of particular importance to ensure effective prevention of obesity, tailored to the needs of children and adolescents.

In the work, the authors attempted to answer the following questions:

1. Are 8 and 16 week of high-intensity interval training (at 80 100% Peak Power Output at ventilation threshold) has a beneficial effect on cardiovascular variables, lipid profile, blood lactate levels in moderately and severely obese boys?

2. How do boys with moderate or severe obesity perceive such an effort?

Currently, we are dealing with a global epidemic of obesity among children and adolescents. In 2019, the World Obesity Federation estimated there would be 206 million children and adolescents aged 5–19 years living with obesity in 2025, and 254 million in 2030. In the study, the authors undertook to assess the effect of 8 and 16-week of high-intensity interval training (at 80-100% of Peak Power Output at the threshold of ventilation) on cardiovascular variables, lipid profile, blood lactate concentration and assessment of perceived exertion in boys with moderate or severe obesity , assuming that boys with a lower body fat content (moderately obese) can improve their cardiovascular parameters, whether after a short or long period of training, while young boys with severe obesity can only improve their lipid profile and insulin resistance. According to the reviewer, the work is original, because undertaking multidirectional research aimed at developing methods of effective combating obesity among children and adolescents is extremely important from the point of view of the prevention of diet-related metabolic diseases and public health.

The strengths of the presented work include both its cognitive value and the possibility of using the obtained results to develop programs for the treatment of obesity and improve body composition and eliminate cardiovascular and metabolic disorders in adolescents with moderate and severe obesity, by dosing the size of the load adapted to the patients and determining the duration of physical activity. It is a pity that the authors did not conduct similar studies among obese girls, which would have allowed to show possible differences depending on gender.

In the discussion, the authors refer to the results of other studies, which show that the HIIT program did not affect the value of lean body mass and fat content of the trunk or abdomen in young overweight and obese women (ref. 50) and did not show a reduction in fat mass in sixty overweight or obese people (ref. 51). How can the different results obtained by the authors of this study be explained?

The conclusions drawn by the authors are correct and result from the analysis of the research results obtained and the contemporary literature on the subject, and constitute an exhaustive answer to the questions contained in the work.

References are well chosen and correctly quoted in the text.

Tables presented in the publication are well constructed, legible and fully take into account the data contained in the content of the work.

Author Response

We uploaded the rebuttal letter

Reviewer 2 Report

Very interesting manuscript, but the introduction should be improved, it is somewhat messy, for the rest of the manuscript I have no comments, it seems to me that the results are clear, as is the discussion.

I only have one question, why only boys and not mixed training?

Author Response

We uploaded the rebuttal letter

Reviewer 3 Report

Introduction

The text provides a comprehensive overview of the potential benefits of different exercise protocols, including long-duration moderate-intensity training and High-Intensity Interval Training (HIIT), on body fat reduction, fat oxidation, insulin resistance, and various cardio-metabolic risk factors. However, it is crucial to connect these general statements to the specific objectives and hypotheses of your study, as well as the characteristics of your study population.

To enhance the clarity and relevance of your manuscript, I recommend expanding on how these general statements apply to your research context and the population under investigation. Specifically, discuss how the findings of previous studies align or differ from what you aim to examine in your study. Consider highlighting the gaps in the existing literature that your research intends to address and emphasize the potential contribution of your study in advancing the current knowledge in the field.

By providing this contextualization and linking the general statements to your specific research objectives and sample characteristics, you will strengthen the scientific rigor and relevance of your manuscript. I encourage you to carefully revise this section to ensure that it aligns closely with the focus of your study and clearly outlines its potential contribution to the field.

The study aims to examine the effects of 8 and 16-week HIIT on cardiovascular variables, lipid profile, blood lactate concentrations, and rating of perceived exertion in boys with moderate or severe obesity. While the study design appears appropriate, I have concerns regarding the novelty and contribution of the research.

The authors state that they hypothesize boys with lower body fat (moderate obesity) may improve their cardiovascular parameters with both short-term and long-term HIIT training, whereas boys with severe obesity may only see improvements in lipid profile and insulin resistance after a sufficient 16-week training period. While these hypotheses provide a framework for the study, they appear to align with what is already known in the literature and do not introduce any innovative or novel objectives.

To strengthen the study's significance and contribution to the field, I recommend that the authors consider revising their objectives to include innovative aspects or new insights. This could involve exploring additional variables, examining potential sex-specific differences, or investigating the impact of HIIT on psychological well-being in this specific population. By doing so, the authors can enhance the originality and value of their research.

Methods

Firstly, it is important to provide more specific information about the recruitment process, including the schools or activities from which the participants were recruited. Additionally, please ensure that you mention obtaining voluntary participation and obtaining informed consent from legal guardians and parents, as ethical considerations are crucial in research involving minors.

Furthermore, it is essential to provide a detailed explanation of how the measurements were conducted, including the instruments used and the protocols followed. For instance, regarding height, body mass (BM), percent body fat (%BF), waist circumference (WC), body mass index (BMI), and lean body mass, it would be beneficial to mention the specific measurement techniques and any quality control procedures implemented.

Moreover, it is important to disclose the sample size calculations that were conducted prior to the study. Including this information helps demonstrate the statistical power of the study and the adequacy of the sample size to detect meaningful effects. Please provide the rationale and method used for determining the sample size, considering the specific variables of interest and the desired effect size.

Lastly, ensure that you mention the procedures followed to maintain the privacy and confidentiality of the participant's data. This includes handling and storing the data securely, anonymizing data during analysis, and adhering to data protection regulations.

I have reviewed the section outlining the training protocol, and while it provides an overview of the interventions, there are some gaps in the description that should be addressed to ensure clarity and reproducibility. I recommend providing additional details as follows:

Specify the exact duration of each training session during the first 4 weeks, the second 4 weeks, and the remaining 8 weeks. Clarify the progressive increase in session duration to ensure a comprehensive understanding of the training program.

Provide information on the rest period between each bout of HIIT during the training sessions. State the duration of the rest periods explicitly, as this information is important for replicating the protocol accurately.

Clarify the specific method used to adjust the exercise intensity every four weeks. Describe the criteria used to determine the need for intensity readjustment and provide detailed instructions on how the adjustment was made (e.g., a percentage increase in power output based on changes in heart rate at the ventilatory threshold).

Describe the standardized breakfast consumed by participants before the tests, including the specific components and quantities. This information will contribute to the standardization of conditions and reduce potential confounding factors.

Please elaborate on the instructions provided to participants regarding their usual physical activity level and diet during the intervention period. Specify any restrictions or recommendations given to ensure consistency among participants.

Additionally, I encourage you to address the dropout of five adolescents during the training period. Provide a brief explanation of the reasons for their dropout and mention how their exclusion was handled in the analysis.

While you have provided an overview of the statistical methods used, there are some gaps in the description that should be addressed to ensure transparency and reproducibility. I recommend providing additional details as follows:

Clarify the specific variables for which normal distribution and homogeneity of variances were assessed using Shapiro-Wilk and Levene's tests, respectively. Specify whether these tests were performed on the baseline characteristics or the outcome variables measured at different time points.

State the specific factors included in the two-way analysis of variance with repeated measures. Clearly indicate the main factors and their levels, as well as the within-group and between-group comparisons made in the analysis.

Provide information on the specific posthoc test used to follow the detection of a significant interaction effect in the two-way analysis of variance. Specify whether Tukey's honestly significant difference test was applied to compare all possible pairs of groups or if it was applied only to the sample's main effect. Clarify the criteria used to determine statistical significance in the post-hoc test.

Elaborate on the calculation of effect sizes (ES) using Cohen's d. Provide a brief explanation of how Cohen's d was computed and its interpretation in the context of your study. State the variables for which effect sizes were calculated and reported.

Explain the rationale for using the Kruskal-Wallis test for between-group changes and clarify which variables were analyzed using this non-parametric test. Specify the posthoc test applied (e.g., Mann-Whitney test) and provide details on how the Bonferroni correction for alpha slippage was applied.

State the version of the Statistical Package for the Social Sciences (SPSS) software used for the analyses, including the release number and the company (IBM) or any other relevant information.

Discussion and Conclusion

I have reviewed the Limitations and Conclusions sections of your study. While you have acknowledged some limitations and drawn conclusions based on the findings, I believe there is room for further exploration of their implications for researchers and physical education teachers. To enhance the practical implications of your study, I recommend the following revisions:

Elaborate on the specific practical implications of the identified limitations. Discuss how these limitations may impact the interpretation and generalizability of your findings and suggest potential avenues for future research to address these limitations. This will provide guidance for researchers who may wish to build upon your study or replicate it with larger and more diverse populations.

Consider the implications of your study's findings for physical education teachers working with obese male adolescents. Discuss how the results of your study can inform the development of exercise programs or interventions in real-world settings. Address considerations such as exercise duration, intensity progression, and exercise tolerance in this specific population. This will provide practical insights for professionals working with obese adolescents in designing effective and safe exercise regimens.

Highlight any specific recommendations or guidelines derived from your study's findings. For example, you could suggest the optimal duration and intensity progression for HIIT programs for moderately and severely obese male adolescents based on your results. This will assist physical education teachers in implementing evidence-based practices and optimizing the outcomes of their interventions.

Emphasize the importance of further research in this field to support and validate your findings. Discuss the potential benefits of conducting additional studies with different populations (e.g., females, different age groups) or incorporating additional variables of interest. This will encourage researchers to explore these areas and contribute to the generalizability and robustness of the evidence base.

The English writing in the manuscript requires revision as there are several grammatical and syntactical errors throughout the text. Additionally, there are instances where clarity and coherence could be improved, making the overall flow of the paper more reader-friendly. A thorough editing process is recommended to ensure the manuscript meets the expected standards of academic writing.

Author Response

We uploaded the rebuttal letter
